# The Double Burden of Isolation and Unemployment: Suicide Risk in Structurally Vulnerable Populations in Japan—A Case Study of Akita Prefecture (2018–2022)

**DOI:** 10.3390/ijerph22091447

**Published:** 2025-09-18

**Authors:** Roseline Yong

**Affiliations:** Department of Community-Based Psychosomatic Health Care, Graduate School of Medicine, Akita University, Akita 010-8543, Japan; roselineyong@med.akita-u.ac.jp

**Keywords:** suicide rates, social isolation, unemployment, living alone, Akita, Japan, rural aging and depopulation, middle-aged men, suicide prevention policy

## Abstract

Suicide in Japan remains elevated and uneven across regions. We hypothesized that (H1) unemployment and (H2) living alone each increase suicide mortality, and that (H3) their combination yields more-than-additive risk, especially among middle-aged men. Using specially tabulated mortality data (2018–2022) from the Japan Suicide Countermeasures Promotion Center, we cross-classified deaths and denominators into 24 strata by sex, age (20–39, 40–59, ≥60), employment (employed/unemployed), and cohabitation (with others/alone). Five-year average rates per 100,000 were computed; between-group differences were tested with chi-square (Holm-adjusted contrasts). Additive interaction between unemployment and living alone was quantified with the Interaction Contrast (ICR) and Synergy Index (SI), and Akita rates were benchmarked against national strata. Prefecture-level quantification and national benchmarking are rarely reported in Japan. Rates differed significantly across employment-by-cohabitation groups in every sex-by-age stratum (*p* < 0.001). Unemployment and living alone each elevated risk, with the highest rate reported among unemployed men aged 40–59 who were living alone (317.1; >14× employed, cohabiting peers at 22.1). Additive interaction was strongest in men aged 40–59 (ICR = 198.3; SI = 3.05) and present in men aged 20–39 and ≥60; among women, interaction was most evident at the ages of 40–59 and sub-additive at ≥60. Compounded effects among men were consistently larger in Akita than nationally, whereas the largest absolute burden fell on unemployed men aged ≥60 who were living with others (203 deaths). The novelty of this investigation lies in quantifying additive interaction with national benchmarking and contrasting per capita risk with absolute burden to guide dual-track prevention. The findings are ecological.

## 1. Introduction

Suicide remains a critical global public health concern, responsible for over 700,000 deaths each year, with many more non-fatal attempts [1]. Risk is particularly heightened among individuals facing social isolation and unemployment—conditions which are increasingly prevalent in aging, economically unstable, and socially fragmented societies [2,3].

Japan exemplifies this paradox. Despite its economic prosperity and universal healthcare system, the country continues to report suicide rates higher than most other Organization for Economic Co-operation and Development (OECD) nations. Although national figures have declined since the early 2000s, suicide remains a leading cause of death—especially among working-age men and older adults [4]. Beneath these national averages lie deep regional disparities, most notably in rural, depopulating areas where demographic aging, economic stagnation, and social fragmentation converge [5].

Akita Prefecture illustrates these structural vulnerabilities. In 2022, its suicide mortality rate stood at 23.7 per 100,000—well above the national average of 16.3 [4]. The region is marked by rapid demographic aging (with over 38% of residents aged 65 or older), sustained youth outmigration, and limited economic diversification [6]. These conditions contribute to widespread solitary living, poor access to mental health services, and restricted employment opportunities—particularly outside of agriculture and small-scale industry. Vulnerability is especially acute among older adults and non-regular workers, who often lack job stability, social protection, and community support [7,8].

Theoretically, unemployment and isolation are established, independent risk factors for suicide—linked to depressive symptoms, suicidal ideation, and diminished social connectedness [9,10]. Among working-age adults—particularly men—these stressors can erode identity, purpose, and integration in settings where masculine identity is strongly tied to stable employment and family roles [11,12].

Joiner’s Interpersonal Theory of Suicide [13] offers a useful framework here, highlighting “thwarted belongingness” and perceived burdensomeness as central psychological mechanisms that may be activated when social roles are lost or unattainable. In Japan’s labor market, over 36% of jobs are non-regular [14], and men in precarious roles show an elevated suicide risk [15]. At the extreme end, consistent social isolation and withdrawal (hikikomori) underscore how chronic withdrawal compounds isolation and risk [16]. Notably, mere participation in work or school does not guarantee emotional connection; individuals with limited social integration may suffer more distress than those who are formally excluded [17]. This underscores the gap between physical inclusion and perceived belonging. Relatedly, higher odds of depressive symptoms among men living alone or in multigenerational households versus spouse-only households was found, with neighborhood social cohesion buffering this risk [18].

Despite extensive research on the individual effects of social isolation and unemployment, their combined impact remains underexplored—particularly among structurally vulnerable populations such as unemployed, middle-aged men living alone. This intersection warrants closer examination.

To address this gap, the present study investigates suicide mortality in Akita Prefecture from 2018 to 2022, analyzing variations by sex, age, employment status, and cohabitation and benchmarking against national data. Specifically, we hypothesize that

**H1.** 
*Unemployment is associated with higher suicide mortality.*


**H2.** 
*Living alone is associated with higher suicide mortality.*


**H3.** 
*The combination of unemployment and living alone produces compounded risk, particularly among middle-aged men.*


By focusing on a demographically and economically marginalized region, this study aims to illuminate how overlapping vulnerabilities interact to elevate suicide risk—and how broader structural forces shape the lived experience of isolation.

## 2. Methods

### 2.1. Data Source and Scope 

This study utilizes specially tabulated suicide mortality data obtained from the Japan Suicide Countermeasures Promotion Center (JSCP), published annually as the Regional Suicide Profile. These profiles are compiled for every prefecture, designated city, and municipality in Japan, and have been distributed since 2017 under the national directive outlined in the 2017 Cabinet-approved General Principles of Suicide Prevention Policy. The aim of the Regional Suicide Profile is to support municipalities in designing and evaluating localized suicide prevention strategies by providing granular data on suicide cases and regional context.

Although the profiles are not publicly disclosed online, they are made available to all local governments upon request and are compiled using officially verified vital statistics (death certificates), municipal registry information, and other administrative records.

The dataset used in this study covers suicide deaths occurring between 2018 and 2022 (Heisei 30 to Reiwa 4) in Akita Prefecture, one of Japan’s most demographically vulnerable regions. For comparative benchmarking, corresponding national data were used from the same JSCP sources.

Importantly, the JSCP aggregates suicide data by demographic attributes and contextual variables, including the following:Sex.Age group.Employment status.Living arrangement (cohabitation status).History of suicide attempts (in some profiles).Means of suicide.Stress, mental health status, and household structure (when available from municipal data).

In small population groups, even one additional suicide case can cause a significant fluctuation in the suicide rate. To minimize year-to-year volatility and better identify structural patterns among different segments of the population, we calculated a five-year average suicide mortality rate. This five-year average was computed based on the total number of suicide deaths and the estimated population in each demographic group, using annual statistics from the Ministry of Internal Affairs and Communications’ Population Estimates.

### 2.2. Variables and Demographic Stratification 

The data were stratified across four variables, yielding a total of 24 demographic subgroups:Sex: Male/Female.Age Group: 20–39, 40–59, ≥60.Employment Status: Employed/Unemployed.Cohabitation Status: Living with others/Living alone.

Each suicide case was assigned to 1 of these 24 strata based on the deceased’s official records. Employment status and living arrangement were determined from municipal and welfare registries or, in some cases, inferred from available records at the time of death registration.

In the JSCP registry, the term “Unemployed” referred to individuals who were not formally registered in paid employment at the time of their death. This classification may include long-term unemployed persons, those engaged in irregular or informal work, and individuals who are outside the labor force for reasons such as retirement, disability, or caregiving. While useful for population-level comparisons, this category is heterogenous and should be interpreted with caution. 

### 2.3. Outcome Measure: Suicide Mortality Risk 

The main outcome variable was the suicide mortality risk per 100,000 population for each demographic stratum. This was calculated using the following formula:(1)$$Suicide\;mortality\;risk=\frac{Number\;of\;suicides\;in\;subgroup}{Population\;of\;subgroup}\times \;100,000$$

Population denominators were taken from Akita Prefecture’s own regional data and national-level comparative groups. National risk rates for each demographic stratum were used as benchmarks, allowing for localized risk amplification to be observed in Akita. To protect confidentiality and minimize disclosure risk in small cells—and to reduce year-to-year volatility in strata with small denominators—we report five-year average rates and do not present annual, stratum-specific rates.

Population denominators and five-year averaging were implemented. The denominators for each sex–age-employment–cohabitation stratum were annual mean population estimates produced by the JSCP using the Ministry of Internal Affairs and Communications’ 2020 Census “employment status” basic tabulations. JSCP apportioned individuals with “unknown” labor-force status across employed and unemployed categories (i.e., employed: “work in addition to housework/study”; unemployed: “unemployed”; plus non-labor-force groups) before deriving annual means. As the outcome is a five-year average (2018–2022), the denominator used for rate calculation is the JSCP one-year mean multiplied by five. Thus,(2)$$Five\text{-}year\;average\;rate=\frac{Deaths\;over\;2018-2022}{\left(JSCP\;annual\;mean\;population\right)\times 5}\times 100,000$$

### 2.4. Analytical Approach 

The analysis was descriptive and comparative in nature. In addition to descriptive comparisons, we conducted simple statistical tests to confirm observed differences. Chi-square tests were applied to compare suicide counts across selected demographic subgroups, and z-tests for two proportions were used to assess differences in suicide mortality rates between Akita and national benchmarks. While the ecological design of the data limits the application of multivariate modeling, these tests provide statistical confirmation of key disparities.

Patterns were also interpreted with reference to the epidemiological concept of causal interaction, meaning that the joint effect of unemployment and solitary living on suicide mortality was greater than the sum of their independent effects. To examine this more explicitly, we calculated two measures of additive interaction [19]: 

The Interaction Contrast (ICR),(3)$$ICR=R_{11}-R_{10}-R_{01}+R_{00}$$
where R_00_ = employed + living with others (reference), R_10_ = employed + living alone, R_01_ = unemployed + living with others, and R_11_ = unemployed + living alone. Interpretation, ICR > 0 → positive interaction (the joint effect is greater than the sum of the individual parts), ICR = 0 → no interaction on the additive scale, and ICR < 0 → antagonism (the joint effect is less than expected).

And the Synergy Index (SI),(4)$$SI=\frac{R_{11}-R_{00}}{{(R}_{10}-R_{00})+(R_{01}-R_{00})}$$

Interpretation: SI = 1 → no interaction (joint effect = sum of individual effects), SI > 1 → positive interaction (joint effect greater than expected), and SI < 1 → antagonist interaction (joint effect smaller than expected). 

Both metrics require the full 2 × 2 cross-classification of employment status (employed vs. unemployed) and cohabitation (living with others vs. living alone), and thus draw on all four corresponding subgroups. For readability, ICR and SI are reported in the table row corresponding to the joint-exposure group (unemployed, living alone), but they are derived from rates across all four exposure categories. 

Special attention was paid to the following:Identifying the highest-risk subgroups.Comparing Akita’s suicide rates for each stratum with national equivalents.Observing gendered and age-specific patterns of compounded vulnerability.

All between-group comparisons and interaction metrics (ICR, SI) used the five-year average rates defined above. The proportions shown in the descriptive tables/figures represent the the total share of deaths within Akita or Japan for 2018–2022; because persons aged <20 years and those in the age-unknown categories are excluded from our 24 strata, the within-table percentages do not sum to 100%.

### 2.5. Ethical Considerations 

All data used in this study were aggregated, publicly available, and devoid of personally identifiable information. As the analysis did not involve human subjects or intervention, ethical approval was not required under institutional or national guidelines.

## 3. Results

Suicide mortality in Akita Prefecture from 2018 to 2022 revealed sharp disparities across demographic lines, shaped by age, sex, employment status, and cohabitation patterns. Table 1 (men) and Table 2 (women) report the counts, denominators, rates, and interaction metrics.

### 3.1. Statistical Confirmation 

Chi-square tests confirmed that suicide rates differed significantly across employment cohabitation groups within every sex-by-age stratum (men 20–39: χ^2^(3) = 73.23, *p* < 0.001; men 40–59; χ^2^(3) = 264.42, *p* < 0.001; men ≥ 60: χ^2^(3) = 128.98, *p* < 0.001; women 20-39, χ^2^(3) = 19.00, *p* < 0.001; women 40–59; χ^2^(3) = 31.42, *p* < 0.001; women ≥ 60: χ^2^(3) = 22.97, *p* < 0.001). Pairwise Holm-adjusted comparisons showed that the unemployed-alone group consistently had significantly higher rates than the employed-with-others reference group in men of all ages (*p* < 0.001) and in middle-aged women (*p* = 0.038); see the significance markers in Table 1 and Table 2.

### 3.2. Independent Effects of Unemployment and Living Alone

Men (Table 1): Holding cohabitation constant, unemployment was associated with markedly higher suicide rates across all age groups. Among men aged 40–59 who were living alone, the rate was 317.1 vs. 59.2 per 100,000 when unemployed versus employed—about a 5-fold increase (+257.9). Elevations were also large in men ≥60 living alone (113.1 vs. 36.7; almost 3-fold, +76.4) and men 30–39 living alone (116.5 vs. 12.2; almost 10-fold, +104.3). Unemployment among cohabiting men likewise raised risk, e.g., 40–59 (81.7 vs. 22.1; almost f-fold, +59.6) and 20–39 (81.0 vs. 22.4; almost 4-fold, +58.6). Significance markers appear in Table 1. 

Considering living arrangement within employment strata, living alone further amplified risk. The contrast was greatest for unemployed men 40–59 (317.1 vs. 81.7; almost 4-fold, +235.4) and unemployed men ≥60 (113.1 vs. 48.6; almost 2-fold, +64.5). Among employed men, living alone was associated with higher rates in 40–59 (59.2 vs. 22.1; almost 3-fold, +37.1) and ≥60 (36.7 vs. 20.5; almost 2-fold, +16.2), while 20–39 showed no elevation (12.2 vs. 22.4).

Women (Table 2): Within the same cohabitation status, unemployment generally conferred higher risk, with especially pronounced differences among women aged 40–59 living alone (45.2 vs. 5.8; almost 8-fold, +39.4) and women aged ≥60 living with others (20.1 vs. 3.1; almost 6-fold, +17.0). Elevations were also evident for women aged 20–39 who were living with others (17.4 vs. 5.1; almost 3-fold, +12.3) and women aged 20–39 living alone (34.6 vs. 21.0; almost 1.7-fold, +13.6). By contrast, among women aged ≥60 living alone, unemployment had little effect (20.2 vs. 19.4).

Examining living arrangements alongside employment strata revealed that living alone was associated with higher rates among employed women aged 20–39 (21.0 vs. 5.1; almost 4-fold, +15.9) and employed women aged ≥60 (19.4 vs. 3.1; almost 6-fold, +16.3). Among unemployed women aged 40–59, living alone also raised the suicide risk (45.2 vs. 15.5; almost 3-fold, +29.7). Differences were minimal for employed women 40–59 (5.8 vs. 5.7) and unemployed women ≥ 60 (20.2 vs. 20.1). 

### 3.3. Joint Effects of Unemployment and Isolation

When unemployment and living alone co-occurred, a more-than-additive joint effect emerged, consistent with the epidemiological concept of causal interaction. Table 1 shows that among men aged 20–39, the ICR was 45.7 per 100,000 and SI = 1.94. The most dramatic synergy occurred among middle-aged men (40–59), with rates rising from 81.7 (unemployed, cohabiting) to 317.1 (unemployed, alone), yielding an ICR of 198.3 and SI = 3.05, indicating that the joint effect was more than three times greater than the sum of individual risks. Men aged ≥60 also showed significant synergy (ICR = 48.3; SI = 2.09). 

Table 2 shows that among women, synergy was most pronounced in the 40–59 age group (ICR = 29.6; SI = 3.99), despite overall lower absolute rates. In contrast, younger women showed little evidence of interaction (ICR = 1.3; SI = 1.05), while older women exhibited sub-additivity (ICR = −16.2; SI = 0.51), suggesting that solitary living added little incremental risk for this group. 

### 3.4. Akita Versus National Comparisons

To examine whether these compounded effects were unique to Akita or reflected broader national trends, interaction metrics were benchmarked against national suicide rates (Table 3). Synergy among men was consistently stronger in Akita than nationally. For instance, among middle-aged men (aged 40–59), the Akita ICR was 198.3 compared to 117.8 nationally, and the SI was 3.05 versus 2.18. This indicates that the compounding disadvantage of unemployment and isolation was markedly more severe in Akita.

Among women, patterns were more heterogeneous. In the 40–59 age group, synergy was stronger in Akita (ICR = 29.6; SI = 3.99) than nationally (ICR = 20.4; SI = 2.21). In contrast, older women (≥60) showed sub-additivity in Akita (ICR = −16.2; SI = 0.51), whereas nationally, the interaction remained modestly positive (ICR = 5.4; SI = 1.58). 

### 3.5. Burden Versus Risk

Table 1 and Table 2 and Figure 1 (counts) show that the largest burden of suicides was recorded for unemployed men aged ≥60 who were living with others (n = 203 deaths over 2018–2022). In contrast, Figure 2 (rates per 100,000) and Table 1 highlight a small but high-risk subgroup: unemployed men aged 40–59 living alone (317.1 per 100,000), who had over fourteen times the reference rate for employed, cohabiting men of the same age (22.1 per 100,000). For women, the burden was concentrated among those aged ≥60 who were unemployed and living with others (n = 152; Table 2), whereas the highest risk appeared among unemployed women aged 40–59 living alone (45.2 per 100,000; Table 2). Together, these complementary views justify dual policy targets: (i) reduce total deaths where numbers are highest, and (ii) mount high-intensity outreach for small, hyper-vulnerable groups. 

## 4. Discussion

Taken together, these findings support H1, partially support H2 (with important sex-and age-specific exceptions), and strongly support H3, with the sharpest compounding among men. What is new here is that we (i) quantify how unemployment and solitary living combine on an additive scale (ICR, SI), (ii) show that this compounding is stronger in Akita than nationally, and (iii) separate high burden from high-risk groups to motivate a dual prevention strategy. These findings reveal stark social gradients in suicide mortality in Akita, shaped by structural precarity and social isolation. The highest relative risk was observed among unemployed, middle-aged men living alone, underscoring the compounding nature of economic exclusion and social living. This pattern reinforces earlier work in Japan linking unemployment to elevated suicide risk [15,16] and highlights how occupational instability continues to erode mental health in contexts where masculine identity is closely tied to employment and family roles.

Living alone also independently heightened risk, especially for men. This finding echoes broader evidence that solitary living can signify social detachment, diminished emotional integration, and limited access to informal care networks—particularly in aging rural communities [9,10]. For women, however, the association between cohabitation and suicide was less consistent. One plausible, but tentative, interpretation is that women may maintain stronger extra-household social ties or face different cultural expectations regarding disclosure and help-seeking [3]. Our ecological data cannot adjudicate mechanisms, and alternative explanations (e.g., reporting differences, cohort effects, health-selection process) should be considered.

Crucially, these structural risks did not operate additively, but interactively. The combination of unemployment and solitary living produced suicide rates far beyond either factor alone. This supports the epidemiological concept of causal interaction, quantified here with the ICR and SI [19]. Middle-aged men displayed especially strong synergy, with joint effects more than three times greater than expected from independent risks. Women aged 40–59 also showed evidence of compounded vulnerability, though older women exhibited sub-additivity, suggesting heterogeneous pathways across the life course. 

These patterns align with Durkheim’s typologies of anomic and egoistic suicide, where weakened social integration and disrupted norms elevate suicidality [20], and also with Joiner’s Interpersonal Theory of Suicide, which emphasizes the role of thwarted belongingness, perceived burdensomeness, and acquired capability for suicide [13]. In the context of Akita, prolonged economic stagnation, demographic aging, and depopulation may collectively foster both social detachment and community-wide exposures to hardship, amplifying vulnerability through structural as well as psychological channels [5,7]. 

These patterns are also consistent with Japan’s phenomenon of hikikomori—prolonged social withdrawal that can persist into midlife. Although our registry data cannot identify hikikomori directly, two features of the results align with this pathway: (i) extreme risk among unemployed, middle-aged men living alone and (ii) a weaker gradient by cohabitation in some strata. Many socially withdrawn adults do not live alone but remain secluded within the parental household; thus, “living with others” can still coincide with severe isolation. In Joiner’s terms, long-term withdrawal may intensify thwarted belongingness and perceived burdensomeness, while unemployment erodes role identity; the elevated ICR and SI among middle-aged men in Akita are consistent with this mechanism [16].

The contrast between high-burden and high-risk groups is particularly instructive. Older, unemployed men living with others accounted for the largest number of suicide deaths, reflecting the cumulative burden of structural vulnerability. In contrast, middle-aged, unemployed men living alone faced the most extreme per capita risk, signaling acute concentration of vulnerability in a smaller subgroup. This duality indicates that prevention policy must simultaneously address the volume of suicide in large groups and the intensity of risk in smaller but hyper-vulnerable ones. 

Compared with prior work that typically examines unemployment or living arrangement in isolation, this study quantifies their additive interaction at population level, benchmarks these interactions against national strata, and pairs rate-based risk with count-based burden to generate actionable, dual-track prevention guidance for structurally vulnerable regions.

When benchmarked nationally, Akita displayed consistently stronger interaction effects among men, particularly in midlife. This suggests that regional structural vulnerabilities—population aging, economic contraction, and outmigration—intensify the psychological costs of unemployment and isolation beyond national averages. Internationally, the compounded risks identified here echo patterns in other East Asian contexts, such as South Korea and Taiwan, where unemployment [21] and solitary living [22] have been linked to sharply elevated suicide rates. At the same time, evidence from European cohorts shows that strong neighborhood social capital can buffer the mental health harms of isolation [23], suggesting that community-level cohesion remains a critical protective factor across settings.

### 4.1. Strengths and Limitations

This study provides a rare ecological analysis quantifying the interaction effects of unemployment and solitary living on suicide mortality. The use of officially verified vital statistics across five years minimized random volatility and allowed for benchmarking against national data. However, several limitations must be acknowledged. First, its ecological design precludes individual-level causal inference, raising the risk of ecological fallacy. Second, the registry category of “unemployed” is heterogeneous, encompassing long-term unemployed, non-regular workers, and those outside the labor force for caregiving or health reasons. Third, cohabitation is an imperfect proxy for social isolation, which is fundamentally subjective and may persist even in multi-person households. This is particularly salient for people experiencing hikikomori, who may be classified as “living with others” despite marked isolation [16]. Fourth, unmeasured confounders such as mental health diagnoses, prior suicide attempts, or access to care were unavailable. Fifth, the exclusion of persons under 20 years limits generalizability to youth suicide patterns. Finally, the cross-sectional design treats 2018–2022 as a single snapshot, masking temporal dynamics—including the impact of the COVID-19 pandemic, which may have reshaped patterns of isolation and employment [4].

### 4.2. Policy and Prevention Implications

Our findings underscore the need for multilevel suicide prevention strategies that address both broad population burden and acute per capita risk, especially in structurally marginalized regions like Akita. The sharp contrast between groups with high suicide counts and those with extreme suicide rates reveals the necessity of dual-focus interventions: on the one hand, population-scale mental health infrastructure for high-burden groups, and on the other, precision-targeted support for acutely vulnerable subpopulations. Because the analysis is ecological, these patterns should guide place-based identification of communities and settings where individual-level screening and support are intensified, rather than stereotyping or universally targeting all persons within a demographic category.

Focused outreach should be targeted toward unemployed, middle-aged men living alone, who face suicide risks exceeding 300 per 100,000—indicating critical levels of structural and social disconnection. However, preventive action cannot stop at this group. Expanding community-based mental health services in rural and depopulating areas is essential, as in these areas, access to care is limited and stigma continues to deter help-seeking. At the same time, strengthening local social cohesion, such as neighborhood mutual aid, community centers, and intergenerational initiatives may buffer the risks of isolation—especially for solitary older adults.

Municipal outreach should include low-threshold, home-based psychosocial services for adults with hikikomori-like withdrawal, with family education, stepwise social re-engagement, and links to employment rehabilitation. These approaches directly address isolation that is not captured by living-alone status and align with the interaction patterns observed in Akita [11,15,16].

Economic interventions also remain critical. Employment reintegration programs that target non-regular and long-term unemployed workers—including skills retraining, job placement, and workplace-based social support—can help restore both material security and a sense of purpose. Complementary to these, early warning systems that draw on municipal and welfare registries could enable proactive engagement with individuals who have recently become unemployed, bereaved, or socially withdrawn, ensuring support reaches those at heightened risk before crises develop. 

Finally, gender-sensitive approaches are necessary. Suicide pathways differ across men and women, shaped by varying social roles, caregiving responsibilities, and access to informal support networks. Interventions must therefore be flexible enough to recognize and respond to these gendered dynamics, rather than assuming uniform vulnerabilities. 

To be effective, suicide prevention policy must move beyond the clinic and address the social, economic, and demographic structures that concentrate risk. Particularly in aging, economically declining prefectures, suicide cannot be separated from broader processes of disconnection, disenfranchisement, and demographic collapse. A structurally informed suicide prevention framework is therefore essential, and should align public health, labor policy, and community development to target both the volume and intensity of suicide risk.

### 4.3. Future Research

Further work should employ longitudinal and individual-level data to clarify causal pathways, including the sequencing of unemployment, isolation, and suicidality. Integrating measures of subjective isolation, social network quality, and neighborhood-level cohesion would allow more precise evaluation of protective and risk factors. Comparative research across regions and countries could also reveal how demographic, cultural, and policy contexts condition the interaction between structural precarity and suicide risk. 

## 5. Conclusions

Consistent with our hypotheses, unemployment increased suicide mortality rate (H1), living alone increased mortality for many—but not for all—groups (H2), and a combination of the two produced the highest risks, particularly among middle-aged men (H3), in Akita Prefecture. In contrast, the largest absolute burden affected older unemployed men living with others, underscoring the need to distinguish per capita risk from case burden when prioritizing action. Effective prevention in aging, depopulating regions should be multi-sectoral—expanding timely access to care, strengthening social integration and employment stability, and coupling system-wide investments with intensive outreach to small, hyper-vulnerable groups at risk of being overlooked. 

## Figures and Tables

**Figure 1 ijerph-22-01447-f001:**
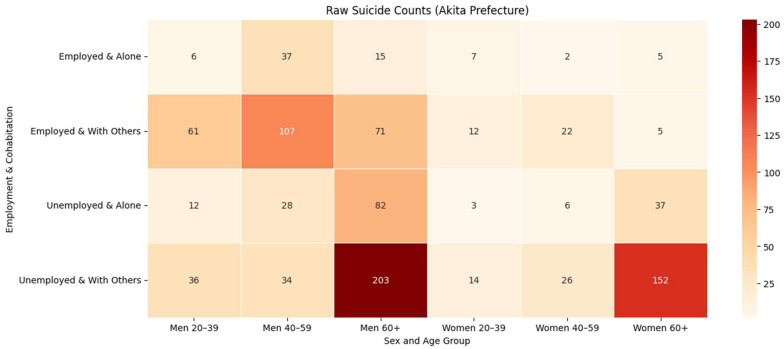
Heatmap of raw suicide counts by demographic group in Akita Prefecture, 2018–2022.

**Figure 2 ijerph-22-01447-f002:**
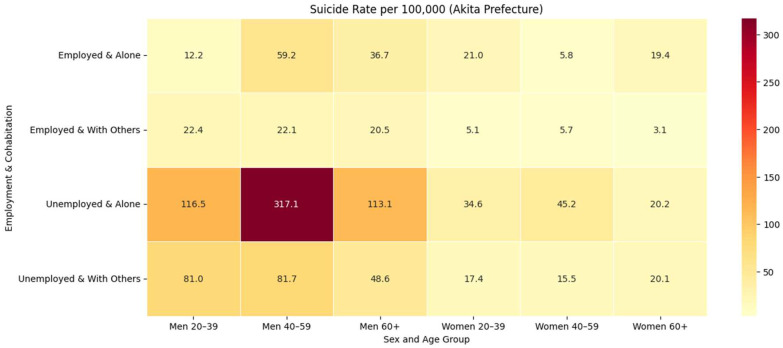
Heatmap of five-year suicide rate per 100,000 by demographic group in Akita Prefecture, 2018–2022.

**Table 1 ijerph-22-01447-t001:** Suicide mortality in men by age, employment, and cohabitation, Akita Prefecture (2018–2022).

Age Group	Employment	Cohabitation	Deaths	Population	Rate (100,000 Population)	Notes (ICR/SI)
20–39	Employed	With others	61	54,378	22.4	Ref group
20–39	Employed	Alone	6	9822	12.2	↓ vs. ref
20–39	Unemployed	With others	36	8884	81.0 (***)	↑ vs. ref
20–39	Unemployed	Alone	12	2060	116.5 (***)	ICR = 45.7; SI = 1.94
40–59	Employed	With others	107	96,845	22.1	Ref group
40–59	Employed	Alone	37	12,492	59.2 (***)	↑ vs. ref
40–59	Unemployed	With others	34	8325	81.7 (***)	↑ vs. ref
40–59	Unemployed	Alone	28	1766	317.1 (***)	ICR = 198.3; SI = 3.05
≥60	Employed	With others	71	69,102	20.5	Ref group
≥60	Employed	Alone	15	8168	36.7	↑ vs. ref
≥60	Unemployed	With others	203	83,619	48.6 (***)	↑ vs. ref
≥60	Unemployed	Alone	82	14,501	113.1 (***)	ICR = 48.3; SI = 2.09

Note: “Ref group” = employed, living with others within the same age stratum. **↑** vs. ref = rate higher than the reference; **↓** vs. ref = rate lower than the reference. Asterisks indicate Holm-adjusted pairwise significance (*** *p* < 0.001); no asterisk = not significant. ICR = Interaction Contrast (per 100,000); SI = Synergy Index—reported for the joint-exposure row (unemployed, alone) and calculated from all four exposure cells.

**Table 2 ijerph-22-01447-t002:** Suicide mortality in women by age, employment, and cohabitation, Akita Prefecture (2018–2022).

Age Group	Employment	Cohabitation	Deaths	Population	Rate (100,000 Population)	Notes (ICR/SI)
20–39	Employed	With others	12	46,735	5.1	Ref group
20–39	Employed	Alone	7	6657	21.0 (*)	↑ vs. ref
20–39	Unemployed	With others	14	16,125	17.4 (*)	↑ vs. ref
20–39	Unemployed	Alone	3	1735	34.6 (*)	ICR = 1.3; SI = 1.05
40–59	Employed	With others	22	77,721	5.7	Ref group
40–59	Employed	Alone	2	6912	5.8	≈ref
40–59	Unemployed	With others	26	33,596	15.5 (**)	↑ vs. ref
40–59	Unemployed	Alone	6	2654	45.2 (***)	ICR = 29.6; SI = 3.99
≥60	Employed	With others	5	32,588	3.1	Ref group
≥60	Employed	Alone	5	5162	19.4 (*)	↑ vs. ref
≥60	Unemployed	With others	152	151,209	20.1 (**)	↑ vs. ref
≥60	Unemployed	Alone	37	36,582	20.2 (**)	ICR = −16.2; SI = 0.51

Note: “Ref group” = employed, living with others within the same age stratum. **↑** vs. ref = rate higher than the reference. Asterisks indicate Holm-adjusted pairwise significance ((* *p* < 0.05; ** *p* < 0.01; *** *p* < 0.001); no asterisk = not significant. ICR = Interaction Contrast (per 100,000); SI = Synergy Index—reported for the joint-exposure row (unemployed, alone) and calculated from all four exposure cells.

**Table 3 ijerph-22-01447-t003:** Interaction metrics (ICR and SI) for suicide mortality in Akita Prefecture versus national benchmarks, 2018–2022.

Sex	Age Group	Akita ICR	Akita SI	National ICR	National SI	Difference (ICR)	Ratio (SI)
Men	20–39	45.7	1.94	26.9	1.57	+18.8	1.24
Men	40–59	198.3	3.05	117.8	2.18	+80.5	1.40
Men	≥60	48.3	2.09	36.7	2.07	+11.6	1.01
Women	20–39	1.3	1.05	13.7	1.86	–12.4	0.56
Women	40–59	29.6	3.99	20.4	2.21	+9.2	1.81
Women	≥60	–16.2	0.51	5.4	1.58	–21.6	0.32

Positive ICR values indicate more-than-additive joint effects of unemployment and solitary living; negative values indicate sub-additivity. Ratios > 1 for SI reflect stronger compounded effects in Akita than nationally.

## Data Availability

The original data presented in the study are openly available in Zenodo at https://shorturl.at/9ekiT.

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
