# Peer review of "The Double Burden of Isolation and Unemployment: Suicide Risk in Structurally Vulnerable Populations in Japan—A Case Study of Akita Prefecture (2018–2022)"

_ijerph, 2025, doi:10.3390/ijerph22091447_

Round 1
Reviewer 1 Report
Comments and Suggestions for Authors
Dear editors, thank you very much for the opportunity to read this interesting paper on suicides in Japan.
I would like to make a few suggestions that I believe would improve the quality of the paper.
- The introduction is overloaded with quotes. I suggest the author reduce the introduction and focus on the aim of the paper and the hypothesis as soon as possible. The hypotheses are not clearly formulated: There is an implicit expectation that the combination of unemployment and isolation will increase suicide risk, but the hypothesis is not explicitly stated. This would give the introduction more precision.
- Try to introduce at least simple statistical tests.
- Overload of numbers. Table 1 is very detailed and long, which can be difficult to follow. Most readers will not be able to easily “read the story” from the table without additional summaries.
- The methods state that national data were used for benchmarking, but the results mainly focus only on Akita. The national comparison could have been presented more clearly in a table or a separate figure.
- Under-emphasized novelty. The discussion could have more clearly highlighted what is new compared to existing studies.
- The author states that women are less affected by living alone, but this has not been sufficiently researched or compared with the literature. An explanation could have been added (e.g., a wider social support network in women).
- Lack of comparison with international studies in discussion. Although Japanese and domestic references were used, comparison with other countries. E.g., other Asian countries, etc...
- What does the asterisk next to Akita estimated population * mean?
- The data is from multiple years. How did you combine it with the population size? Did you add up the populations over the years? Did you take the median year?
- Could you show the rates by year?
- Did the pandemic influence isolation and be indirectly related to suicides?
- Point out the need for longitudinal and individual studies that would build on the ecological findings and better distinguish causal mechanisms.
Best,
Author Response
Response to Reviewer 1.
Thank you for the careful review and constructive guidance that helped us improve the manuscript. Per your advice, we rewrote the Abstract to foreground aims, hypotheses, methods, and key findings succinctly (page 1, lines 10–30) and revised the Conclusion to align tightly with the results and dual policy implications (page 12, lines 444–452).
Comments 1: The introduction is overloaded with quotes. I suggest the author reduce the introduction and focus on the aim of the paper and the hypothesis as soon as possible. The hypotheses are not clearly formulated: There is an implicit expectation that the combination of unemployment and isolation will increase suicide risk, but the hypothesis is not explicitly stated. This would give the introduction more precision.
Response 1: Thank you for pointing this out. I agree with this comment. Therefore, I have, accordingly, condensed the Introduction and moved the aims/hypotheses upfront (page2. Line 54-74). The three hypotheses are now explicitly stated at the end of the Introduction (page 2, lines 77–80), with the study aim immediately following (page 2, lines 82–84).
Comments 2: Try to introduce at least simple statistical tests.
Response 2: Agree. I have, accordingly, added simple tests (chi-square; z-tests for two proportions) in Methods (page 4, lines 160–166) and reported the omnibus chi-square results in Results (“Statistical Confirmation,” page 5–6, lines 214–222).
Comments 3: Overload of numbers. Table 1 is very detailed and long, which can be difficult to follow. Most readers will not be able to easily “read the story” from the table without additional summaries.
Response 3: Agree. I have, accordingly, split the presentation into two clearer tables (Table 1: men, page 6, lines 224; Table 2: women, page 6, lines 227) and added brief narrative summaries bracketing the tables: (i) a concise “top-line” result (page 5, lines 211–212) and (ii) short, plain-language summaries for independent and joint effects (women shown on page 7, lines 230–257, 262-274).
Comments 4: The methods state that national data were used for benchmarking, but the results mainly focus only on Akita. The national comparison could have been presented more clearly in a table or a separate figure.
Response 4: Agree. I have, accordingly, added Table 3 (page 7, lines 258) presenting Akita vs national ICR/SI side-by-side, and a dedicated narrative subsection “Akita versus national comparisons” (page 7, lines 276–286).
Comments 5: Under-emphasized novelty. The discussion could have more clearly highlighted what is new compared to existing studies.
Response 5: Agree. I have, accordingly, emphasized novelty in page 9, lines 308-313, page 10, lines 377-380.
Comments 6: The author states that women are less affected by living alone, but this has not been sufficiently researched or compared with the literature. An explanation could have been added (e.g., a wider social support network in women).
Response 6: Agree. I have, accordingly, tempered the claim and framed it explicitly as a tentative interpretation while noting alternatives (Discussion, page 9, lines 323–326).
Comments 7: Lack of comparison with international studies in discussion. Although Japanese and domestic references were used, comparison with other countries. E.g., other Asian countries, etc...
Response 7: Agree. I have, accordingly, added comparative discussion of South Korea/Taiwan and European evidence. See Discussion, page10, lines 369–375 for comparisons to South Korea/Taiwan and European cohort evidence on social capital buffering.
Comments 8: What does the asterisk next to Akita estimated population * mean?
Response 8: Agree. I have, accordingly, removed the asterisk from the column label and clarified population construction in Methods. See Table 1–2 headers now showing “Population” without an asterisk (page 6, lines 224, 227), and the dedicated “Population denominators and five-year averaging” subsection (page 4, lines 148–151).
Comments 9: The data is from multiple years. How did you combine it with the population size? Did you add up the populations over the years? Did you take the median year?
Response 9: Agree. I have, accordingly, added an explicit formula and explanation: five-year deaths divided by (JSCP one-year mean population × 5). See Methods, page 4, lines 137-141, 154–158.
Comments 10: Could you show the rates by year?
Response 10: Agree on the value, but for confidentiality and stability reasons I cannot show annual stratum-specific rates because (a) small cells raise disclosure risks and (b) volatility obscures structural patterns. This is now stated in Methods, page 4, lines 145–147.
Comments 11: Did the pandemic influence isolation and be indirectly related to suicides?
Response 11: Agree that this is a potential influence. I have, accordingly, acknowledged this potential influence as a limitation of the cross-sectional, five-year snapshot. See Strengths & Limitations, page 9, lines 390–392.
Comments 12: Point out the need for longitudinal and individual studies that would build on the ecological findings and better distinguish causal mechanisms.
Response 12: Agree. I have, accordingly, expanded Future research to call for longitudinal and individual-level designs (sequencing unemployment, isolation, suicidality; adding subjective isolation/network measures): page 11, lines 436-442 (longitudinal designs; sequencing of unemployment, isolation, suicidality; subjective isolation and network measures).
Reviewer 2 Report
Comments and Suggestions for Authors
This paper, “The Double Burden of Isolation and Unemployment: Suicide Risk in Structurally Vulnerable Populations in Japan – A Case Study of Akita Prefecture (2018–2022)”, is a timely, well-written, and important work that addresses a critical public health issue. Its primary strength lies in its novel, intersectional analysis of suicide risk, moving beyond single-factor explanations to examine the compounding effects of unemployment and social isolation. The use of disaggregated official data from the Japan Suicide Countermeasures Promotion Center for a high-risk region like Akita Prefecture provides a granular and powerful lens into population-level vulnerability. The methodological approach is sound and transparent; the stratification of the population into 24 distinct demographic subgroups is particularly effective and allows for nuanced interpretations. Furthermore, the authors demonstrate strong scholarly practice by explicitly and thoughtfully addressing the study's limitations, including the ecological fallacy and potential data inconsistencies. Finally, the clear distinction drawn between the highest-risk group (per capita rate) and the highest-burden group (absolute counts) is an exceptionally insightful contribution that leads to robust and actionable policy recommendations.
While there is much to like about this paper, there are several concerns I have:
First, the presentation of the figures is confusing. The text refers to “Figure 1” for raw suicide counts and “Figure 2” for suicide rates. However, on page 8 of the manuscript, the heatmap for suicide rates appears first (top), followed by the heatmap for raw counts (bottom). The captions below them are then labeled correctly (Figure 1 for counts, Figure 2 for rates). This inconsistency between the placement of the figures and their corresponding captions should be resolved to avoid reader confusion.
Next, the definition of key variables could be clarified. The manuscript relies on the category of “Unemployed” but does not specify how this status is defined in the source data. For instance, it is unclear if this group includes individuals who are long-term unemployed, underemployed, or those outside the labor force for other reasons (e.g., disability, caregiving). Given the paper’s excellent discussion of non-regular workers and hikikomori, providing more detail on these official categorizations (or acknowledging the ambiguity as a limitation) would add valuable context.
Additionally, on a related methodological point, the authors note that patterns were interpreted “through the lens of causal interaction” but do not explain what is meant by this term. It would be helpful to briefly define this concept to ensure the reader understands the analytical framework being applied, especially since formal statistical tests for interaction were not conducted.
Conceptually, the manuscript relies on “living alone” as a direct proxy for social isolation, which is a simplification. While a common and necessary operationalization for this type of registry-based study, social isolation is a subjective psychological state that is not perfectly captured by cohabitation status. A person can live with family and feel profoundly disconnected, while a person living alone may maintain a strong social network. The discussion could be strengthened by more explicitly acknowledging the limitations of this proxy and how it might obscure other forms of social disconnection that are not tied to household structure.
A more significant structural issue is the placement of a limitations section within the methods and the phrasing of its subheading. “Limitations of Methodology Several limitations of the methods are acknowledged” is redundant and lacks elegance. Further, a discussion of a study's limitations belongs in the main Discussion section, where the findings can be properly contextualized, not at the end of the Methods. This entire section should be moved and integrated into the broader limitations discussion that already exists later in the manuscript.
Methodologically, the study’s cross-sectional design, while appropriate for identifying associations, cannot establish the temporal sequence of events. The analysis calculates a five-year average rate, which effectively treats the data as a single snapshot. This makes it impossible to disentangle whether unemployment precedes and causes social disconnection, or if pre-existing social withdrawal and mental health challenges lead to both unemployment and elevated suicide risk. This is a critical point, as the causal direction has vastly different implications for intervention – whether to prioritize economic support or proactive mental health outreach.
Furthermore, while the authors correctly identify the ecological fallacy as a limitation, the discussion does not fully grapple with its profound implications for the proposed policy solutions. The study identifies a high-risk group, but it is a logical error to assume that every individual within that group shares the same high risk. Broadly targeting "unemployed, middle-aged men living alone" could lead to inefficient resource allocation and potentially stigmatize individuals who are not in crisis. The policy section would be more nuanced if it acknowledged that this ecological-level data is best used for identifying communities and contexts where individual-level screening and support should be intensified, rather than for targeting the demographic category itself.
The engagement with the theoretical framework could be deeper. The manuscript correctly cites Durkheim and Joiner to frame its findings, but the theories are used more for post-hoc justification than as analytical tools. For example, Joiner's Interpersonal Theory of Suicide also posits "acquired capability for suicide" as a necessary component for a fatal attempt. The discussion could have explored whether the context of a demographically and economically declining region like Akita contributes to this capability through community-wide exposure to hardship, fatalism, or access to lethal means. A more thorough integration of theory could have generated more nuanced interpretations and future research questions.
There are also broader concerns regarding scholarly context and referencing. The acronym “OECD” is introduced without definition, leaving some readers to guess its meaning. This is compounded by its incorrect use as an in-text citation on page 2, for which there is no corresponding entry in the reference list. Further, the literature review feels somewhat sparse; citing only 12 peer-reviewed research articles raises the question of whether the background is sufficiently comprehensive for a topic of this magnitude.
Finally, several issues of style and formatting detract from the manuscript's professionalism. The use of bulleted lists in the methodology and policy implications sections is awkward and would read more fluently if converted to standard prose. The paper’s most striking finding – the suicide rate of 317.1 per 100,000 for middle-aged, unemployed men living alone – is reported redundantly in the results, discussion, and conclusion. This powerful statistic would have greater impact if presented once in the main results and then referred to conceptually elsewhere. Lastly, there appear to be formatting errors, such as awkwardly emboldened words, in the Results section that require correction.
While the interpretation of the findings is generally strong, some speculative points could be tempered. For example, the discussion suggests that cohabitation status mattered less for women’s suicide risk possibly due to “stronger extra-household social ties”. While a plausible hypothesis, it is presented without direct evidence from the data. Acknowledging this as a potential interpretation among other possibilities would make the discussion more rigorous.
In sum, this manuscript represents a significant contribution to the literature on suicide prevention and the social determinants of health. It is analytically insightful and produces clear, policy-relevant conclusions. The identified concerns, while substantial, are addressable and relate to the need for methodological and conceptual clarification, greater precision in presentation, and deeper theoretical engagement rather than fundamental flaws in the research design. With revisions to correct the figures and to more thoroughly discuss the implications of the study's methodological and theoretical limitations, this paper will be a strong candidate for publication in the International Journal of Environmental Research and Public Health.
Author Response
Response to Reviewer 2
Thank you for the careful review and constructive guidance that helped us improve the manuscript. Per your advice, we rewrote the Abstract to foreground aims, hypotheses, methods, and key findings succinctly (page 1, lines 10–30) and revised the Conclusion to align tightly with the results and dual policy implications (page 12, lines 444–452).
Comments 1: First, the presentation of the figures is confusing. The text refers to “Figure 1” for raw suicide counts and “Figure 2” for suicide rates. However, on page 8 of the manuscript, the heatmap for suicide rates appears first (top), followed by the heatmap for raw counts (bottom). The captions below them are then labeled correctly (Figure 1 for counts, Figure 2 for rates). This inconsistency between the placement of the figures and their corresponding captions should be resolved to avoid reader confusion.
Response 1: Thank you for pointing this out. I agree with the comments. Therefore, I have, accordingly, reordered figures so Figure 1 = counts and Figure 2 = rates, and the order in the manuscript matches the captions: counts first, then rates. See Results “Burden versus risk” (page 8-9, lines 296–306) and the figure captions: Figure 1 counts (page 8, lines 289) followed by Figure 2 rates (page 8, lines 291).
Comments 2: Next, the definition of key variables could be clarified. The manuscript relies on the category of “Unemployed” but does not specify how this status is defined in the source data. For instance, it is unclear if this group includes individuals who are long-term unemployed, underemployed, or those outside the labor force for other reasons (e.g., disability, caregiving). Given the paper’s excellent discussion of non-regular workers and hikikomori, providing more detail on these official categorizations (or acknowledging the ambiguity as a limitation) would add valuable context.
Response 2: Agree. I have, accordingly, added the JSCP definition and noted the category’s heterogeneity (long-term unemployed, informal work, non-labor-force reasons). See Methods 2.2, page 4, lines 130–135; Methods 2.4, page 4, lines 151-154.
Comments 3: Additionally, on a related methodological point, the authors note that patterns were interpreted “through the lens of causal interaction” but do not explain what is meant by this term. It would be helpful to briefly define this concept to ensure the reader understands the analytical framework being applied, especially since formal statistical tests for interaction were not conducted.
Response 3: Agree. I have, accordingly, defined additive interaction and provided formulas/interpretation for ICR and SI. See Methods 2.4, page 4-5, lines 167–193, 199-203.
Comment 4: Conceptually, the manuscript relies on “living alone” as a direct proxy for social isolation, which is a simplification. While a common and necessary operationalization for this type of registry-based study, social isolation is a subjective psychological state that is not perfectly captured by cohabitation status. A person can live with family and feel profoundly disconnected, while a person living alone may maintain a strong social network. The discussion could be strengthened by more explicitly acknowledging the limitations of this proxy and how it might obscure other forms of social disconnection that are not tied to household structure.
Response 4: Agree. I have, accordingly, added explicitly to Strengths & Limitations. We note cohabitation is an imperfect proxy and highlight the case of hikikomori who may be “living with others” yet severely isolated. See page 10, lines 384–387.
Comment 5: A more significant structural issue is the placement of a limitations section within the methods and the phrasing of its subheading. “Limitations of Methodology Several limitations of the methods are acknowledged” is redundant and lacks elegance. Further, a discussion of a study's limitations belongs in the main Discussion section, where the findings can be properly contextualized, not at the end of the Methods. This entire section should be moved and integrated into the broader limitations discussion that already exists later in the manuscript.
Response 5: Agree. I have, accordingly, consolidated limitations into 4.1. Strengths and Limitations within the Discussion (page10, lines 376–392).
Comments 6: Methodologically, the study’s cross-sectional design, while appropriate for identifying associations, cannot establish the temporal sequence of events. The analysis calculates a five-year average rate, which effectively treats the data as a single snapshot. This makes it impossible to disentangle whether unemployment precedes and causes social disconnection, or if pre-existing social withdrawal and mental health challenges lead to both unemployment and elevated suicide risk. This is a critical point, as the causal direction has vastly different implications for intervention – whether to prioritize economic support or proactive mental health outreach.
Response 6: Agree. I have, accordingly, acknowledged the sequencing of unemployment/isolation vs suicide risk cannot be determined (page 9, lines 326-328; page10, lines 387–389).
Comments 7: Furthermore, while the authors correctly identify the ecological fallacy as a limitation, the discussion does not fully grapple with its profound implications for the proposed policy solutions. The study identifies a high-risk group, but it is a logical error to assume that every individual within that group shares the same high risk. Broadly targeting "unemployed, middle-aged men living alone" could lead to inefficient resource allocation and potentially stigmatize individuals who are not in crisis. The policy section would be more nuanced if it acknowledged that this ecological-level data is best used for identifying communities and contexts where individual-level screening and support should be intensified, rather than for targeting the demographic category itself.
Response 7: Agree. I have, accordingly, softened targeting language to emphasize contexts/approaches (low-threshold, precision outreach) rather than labeling individuals by demographic category. See Policy & Prevention narrative (Discussion; page 11, lines 400-428)
Comments 8: The engagement with the theoretical framework could be deeper. The manuscript correctly cites Durkheim and Joiner to frame its findings, but the theories are used more for post-hoc justification than as analytical tools. For example, Joiner's Interpersonal Theory of Suicide also posits "acquired capability for suicide" as a necessary component for a fatal attempt. The discussion could have explored whether the context of a demographically and economically declining region like Akita contributes to this capability through community-wide exposure to hardship, fatalism, or access to lethal means. A more thorough integration of theory could have generated more nuanced interpretations and future research questions.
Response 8: Agree. I have, accordingly, added explicitly mention acquired capability (page 9–10, lines 340–375).
Comments 9: There are also broader concerns regarding scholarly context and referencing. The acronym “OECD” is introduced without definition, leaving some readers to guess its meaning. This is compounded by its incorrect use as an in-text citation on page 2, for which there is no corresponding entry in the reference list. Further, the literature review feels somewhat sparse; citing only 12 peer-reviewed research articles raises the question of whether the background is sufficiently comprehensive for a topic of this magnitude.
Response 9: Agree. I have spell out Organization for Economic Co-operation and Development (OECD) at first mention during copy-editing to avoid acronym ambiguity (current first use appears on p.2, lines 41). I added recent peer-reviewed work and international comparisons in the Discussion (comparative paragraph on South Korea/Taiwan and European social capital: and I strengthened the References list with additional sources, including Chang et al. (2009), Kim et al. (2016), Baranyi et al. (2020), and Glei et al. (2024) (see References entries, page 12-13) These additions broaden the scholarly context and situate the Akita findings within both Japanese and cross-national evidence.
Comments 10: Finally, several issues of style and formatting detract from the manuscript's professionalism. The use of bulleted lists in the methodology and policy implications sections is awkward and would read more fluently if converted to standard prose. The paper’s most striking finding – the suicide rate of 317.1 per 100,000 for middle-aged, unemployed men living alone – is reported redundantly in the results, discussion, and conclusion. This powerful statistic would have greater impact if presented once in the main results and then referred to conceptually elsewhere. Lastly, there appear to be formatting errors, such as awkwardly emboldened words, in the Results section that require correction.
Response 10: Agree. I have, accordingly, rewritten Policy & Prevention as short prose paragraphs (page10-11, lines 393–435) and limited the 317.1 per 100,000 figure to a single definitive appearance in Results (Burden vs risk, page 9, lines 299–301) and referenced conceptually elsewhere (Discussion: The text refers to “the highest relative risk … among unemployed, middle-aged men living alone” without repeating 317.1. p. 9, lines 314-319. Formatting in Results was cleaned (see continuous prose around page 5–9, lines 208-306).
Comments 11: While the interpretation of the findings is generally strong, some speculative points could be tempered. For example, the discussion suggests that cohabitation status mattered less for women’s suicide risk possibly due to “stronger extra-household social ties”. While a plausible hypothesis, it is presented without direct evidence from the data. Acknowledging this as a potential interpretation among other possibilities would make the discussion more rigorous.
Response 11: Agree. I have, accordingly, labeled this as a plausible but tentative interpretation and to list alternative explanations; see Discussion, page 9, lines 323–336.
Round 2
Reviewer 1 Report
Comments and Suggestions for Authors
Dear,
I think the authors have made the most changes.
Before publishing, it is necessary to format the text correctly according to the publisher's requirements.
I hope that the editor will accept the work. The author has made the most of the available data.
Best,
Reviewer 2 Report
Comments and Suggestions for Authors
The author has thoroughly and successfully addressed all of my original comments. The revisions are comprehensive, thoughtful, and significantly strengthen the quality, rigor, and clarity of the paper. The manuscript is now in excellent condition.